# Epidemiology and Drug Resistance Patterns of *Mycobacterium tuberculosis* in High-Burden Area in Western Siberia, Russia

**DOI:** 10.3390/microorganisms11020425

**Published:** 2023-02-08

**Authors:** Irina Kostyukova, Oksana Pasechnik, Igor Mokrousov

**Affiliations:** 1Bacteriology Laboratory, Clinical Anti-Tuberculosis Dispensary, Omsk 644058, Russia; 2Department of Public Health, Omsk State Medical University, Omsk 644050, Russia; 3Laboratory of Molecular Epidemiology and Evolutionary Genetics, St. Petersburg Pasteur Institute, St. Petersburg 197101, Russia; 4Henan International Joint Laboratory of Children’s Infectious Diseases, Children’s Hospital Affiliated to Zhengzhou University, Zhengzhou 450018, China

**Keywords:** tuberculosis, incidence, prevalence, drug resistance, *Mycobacterium tuberculosis*

## Abstract

Russia is a high-burden area for multidrug-resistant tuberculosis (MDR-TB). Here, we studied the epidemiological situation and drug resistance patterns of *Mycobacterium tuberculosis* in the Omsk region in Western Siberia. *M. tuberculosis* isolates (*n* = 851) were recovered from newly diagnosed TB patients in 2021. The isolates were tested by bacteriological and molecular methods, and long-term epidemiological data were analyzed. The TB incidence dec, this is not variablereased from 93.9 in 2012 to 48.1 in 2021, per 100,000 population, but the primary MDR-TB rate increased from 19.2% to 26.4%. The destructive forms of tuberculosis accounted for 37.8% of all cases, while 35.5% of patients were smear-positive. Of all isolates tested, 55.2% were culture-positive, of which 94.5% were further tested for phenotypic drug resistance and associated mutations. More than half (53.4%) of isolates were drug-resistant, 13.9% were monoresistant and 67.9% were MDR. Among MDR isolates, 40.4% were pre-XDR, and 19.2% were XDR. The spectrum of drug resistance included second-line drugs (new-generation fluoroquinolones, linezolid), which significantly increase the risk of an adverse outcome in patients. In conclusion, our results highlight the critical importance of monitoring drug resistance in circulating *M. tuberculosis* strains emerging due to ineffective treatment and active transmission.

## 1. Introduction

The World Health Organization has developed the End TB Strategy, which set the most ambitious goals of reducing the incidence of tuberculosis (TB) by 90% compared to 2015 and deaths due to TB by 95% by 2035, as well as eliminating the economic and social burden associated with this disease [1,2]. At the same time, the achievement of the goals set in the fight against TB is threatened by TB/HIV coinfection, the spread of drug-resistant *Mycobacterium tuberculosis* strains, and finally, the most recent COVID-19 pandemic, which places a significant burden on the global economy [3,4,5]. In 2021, the WHO estimated that 10.6 million people (95% CI: 9.9–11 million) developed TB worldwide, which is equivalent to an incidence of 134 (95% CI: 125–143) per 100,000 population. In 2021, the global TB incidence increased by 4.5%, reversing a long-term trend of a moderate 2% annual decrease over the past decade [6].

In 2006, the WHO defined extensive drug resistance in *M. tuberculosis* as resistance to isoniazid and rifampicin, one of the fluoroquinolones, and at least one of the three second-line injectable drugs kanamycin, capreomycin, or amikacin. In 2020, given the increasing use of new anti-TB drugs (bedaquiline, linezolid, delamanid) and reduced use of some of the injectable drugs, the new XDR and pre-XDR definitions were adopted. The updated WHO guidelines for TB treatment characterize extensive drug resistance as resistance to rifampicin with or without resistance to isoniazid, in combination with resistance to any fluoroquinolone (pre-XDR) and at least linezolid or bedaquiline (XDR) [7,8].

Since 1994, the WHO has carried out global surveillance of resistance of circulating *M. tuberculosis* to first- and second-line drugs and their combinations, including resistance to rifampin and isoniazid [9]. In 2022, new methods for assessing the global distribution of multidrug-resistant (MDR) strains based on the rifampin resistance (RR) were developed [10].

The number of cases of tuberculosis caused by strains resistant to rifampicin at the global level was estimated at 465,000 (95% CI: 400,000–535,000), with about half of the estimated cases detected in three countries—India, China, and Russia [11]. Globally, approximately 2.9% (2.2–3.8%) of newly diagnosed and 17% (9.2–27%) of previously treated cases of pulmonary TB in 2020 were infected by MDR *M. tuberculosis* strains [12].

The countries with the largest share of incident cases of MDR/RR-TB in 2021 were India (26% of global cases), Russia (8.5% of global cases), and Pakistan (7.9% of global cases). Globally, the estimated proportion of pre-XDR cases in 2021 was 20% (95% CI: 16–26%) [6]. In terms of MDR-TB prevalence, Russia ranks third, with extensively drug-resistant tuberculosis (XDR) being increasingly registered; however, data on the spread of XDR-TB are limited even for XDR defined according to the previous WHO guidelines [13,14,15].

An increased prevalence of MDR-TB was linked to the locally predominant endemic or epidemic clusters of *M. tuberculosis* [16,17]. The emergence of MDR and XDR-TB was also influenced by global challenges, such as wars, local armed conflicts, migration, deprivation, and social conflicts, that together threaten the progress in the fight against TB [18,19]. In addition, totally resistant strains, that is, strains resistant to all known anti-tuberculosis drugs, have been discovered in recent years [20]. In 2009, the term TDR-TB was proposed to describe strains resistant to all tested drugs [21,22]. However, this terminology change was not adopted by the WHO because of a lack of a clear and universal approach to defining such strains globally.

TB and MDR-TB are not evenly prevalent across the large territory of the Russian Federation (Figure 1). The Asian part of the country, namely Siberia and the Far East, is marked with the highest TB incidence, prevalence, and mortality aggravated by TB/HIV coinfection. In Siberia, the incidence of HIV infection was 65.5/100,000, and the incidence of HIV-TB coinfection was 18.8/100,000. The MDR-TB incidence in Siberia exceeded the average Russian level by 1.4–1.7 times [23,24].

TB incidence is unequally distributed across the country, and the Siberian Federal District is characterized by a high prevalence of TB infection, morbidity, and mortality [25]. The TB/HIV prevalence in 2019 in Siberia was the highest among Russian Federal Districts (46.4/100,000) and 2.3 higher than the national average (Figure 1) [26]. The Omsk region has a population of 1.85 million people and it is the 6th most populated region in Siberia and 28th most populated in Russia. TB incidence decreased in the Omsk region to 66.5 per 100,000 in 2019 (compared to 89.3 in 2013). However, the prevalence of XDR-TB increased from 5.0 in 2011 to 13.6 per 100,000 in 2017, and the incidence of XDR-TB was 1.6 per 100,000 in 2017 [27]. A particularly adverse feature of this Russian region is the very high rate of TB/HIV coinfection. Its incidence increased from 0.3 in 2006 to 17.2 per 100,000 in 2019, and its prevalence increased from 12.8 to 29.4 per 100,000 population [28].

This study aimed to characterize the epidemiological situation and patterns of drug resistance of *M. tuberculosis* strains isolated from newly diagnosed patients with pulmonary TB in the high-TB-burden area of the Omsk province in the southern part of Western Siberia, Russia.

## 2. Materials and Methods

### 2.1. Study Setting

The study was conducted in the Omsk region in the south of Western Siberia. The region is located on the Russia–Kazakhstan border (Figure 1). The study included 851 patients who were first diagnosed in 2021.

Institutional Review Board Statement: This study was approved by the Ethical Committee of the Omsk State Medical University (protocol # 4 of 14 January 2022). All patients gave informed written consent to participate in this study.

Clinical diagnostic and laboratory examination of patients was carried out in accordance with national clinical guidelines and protocols [8] in 32 local medical organizations in different districts of the province, including the primary collection of sputum for further microbiological examination. Laboratory diagnosis of tuberculosis was carried out according to the same algorithm. Microbiological studies were carried out in the central bacteriology laboratory at the clinical TB dispensary in the city of Omsk.

### 2.2. M. tuberculosis Culture and Species Identification

The diagnostic material was pre-treated with N-acetyl-l-cysteine in combination with 1% NaOH (NALC–NaOH) and inoculated into Loewenstein–Jensen (LJ) and Finn II egg-based media as recommended in previous works [29,30,31]. Primary identification of mycobacteria was carried out based on visual registration of the culture’s growth (not earlier than 3–4 weeks of incubation) and its features (colonies of characteristic morphology and color). The primary identification of strains included microscopic confirmation of the Ziehl–Neelsen-stained bacteria and assessment of the cord factor.

An immunochromatographic test was used to detect a fraction of the mycobacterial protein MPT64, which is isolated from MBT cells during cultivation on liquid and solid nutrient media [25]. SD BIOLINE TB Ag MPT64 Rapid tests (Standard Diagnostics, Korea) were used for identification. The test was applied to both samples cultured on solid and liquid nutrient media. The presence of two colored bands “T” and “C” in the result window indicated a positive result.

To differentiate individual species within the *M. tuberculosis* complex, we used the main biochemical tests to determine nicotinic acid and nitrate reductase, and an additional test for colony growth on nutrient media containing thiophene-2-carboxylic acid hydrazide (2 μg/mL). Unlike *M. tuberculosis*, only *M. bovis* is sensitive to low concentrations of thiophene-2-carboxylic acid hydrazide and is characterized by negative results of biochemical tests. No *M. bovis* isolates were detected in this study.

To differentiate *M. tuberculosis* from non-tuberculous acid-fast mycobacteria, the following basic biochemical tests were used: nitrate reductase test, thermostable catalase test, salicylic sodium test, and niacin test. We additionally assessed the growth on nutrient media containing Tween 80 and thiophene-2-carboxylic acid hydrazide.

### 2.3. M. tuberculosis Drug Susceptibility Testing

Following WHO recommendations, the following culture methods were used to determine drug resistance: a modified proportion method on a liquid nutrient medium in a system with automatic growth detection (Bactec MGIT 960 (Becton Dickinson, Sparks, NV, USA)) for first-line drugs (streptomycin, isoniazid, rifampicin, ethambutol, pyrazinamide) and second-line drugs (levofloxacin, moxifloxacin, amikacin, linezolid) and by the method of absolute concentrations on a solid LJ medium for first-line drugs (streptomycin, isoniazid, rifampicin, ethambutol). For the second-line drugs, the absolute concentration method has not been validated but was approved for use [32].

Critical concentrations for the proportion method in the Bactec MGIT 960 system were as follows: isoniazid—0.1 mg/L, rifampicin—0.5 mg/L, ethambutol—5.0 mg/L, pyrazinamide—100 mg/L, streptomycin—1.0 mg/L, levofloxacin—1.0 mg/L, moxifloxacin—0.25 mg/L, linezolid—1.0 mg/L, ethambutol—5.0 mg/L, and amikacin—1.0 mg/L. Critical concentrations for the method of absolute concentrations were as follows: isoniazid—1 mg/L, rifampicin—40 mg/L, ethambutol—2 mg/L, streptomycin—10 mg/L, ofloxacin—2 mg/L, capreomycin—30 mg/L, and kanamycin—30 mg/L [29].

### 2.4. Molecular Methods

Sputum or bronchoalveolar lavage samples from patients with respiratory tuberculosis were used, regardless of the results of microscopy of these samples. Amplitub assay reagents (Sintol, Moscow, Russia) were used for sample preparation and PCR. The sample preparation included the inactivation of mycobacteria, liquefaction, and homogenization of respiratory samples using the Amplitube-Prep inactivating reagent. Mycobacterial DNA was extracted using an M-Sorb-Tub kit. The kit provides cell lysis, DNA adsorption on magnetic particles, DNA precipitation by centrifugation, DNA washing, and elution. An internal positive control was added to each test sample, to assess the presence or absence of PCR inhibitors and the efficiency of DNA extraction. Real-time PCR for the detection of the *M. tuberculosis* complex was carried out using the Amplitub-RV assay based on the simultaneous detection of two specific *M. tuberculosis* complex genes IS6110 and regX3.

Genotypic determination of antibiotic resistance of *M. tuberculosis* to first- and second-line drugs was carried out using Amplitub-MDR-RV (for isoniazid and rifampicin) and Amplitub-FQ-RV (for fluoroquinolones) kits (Syntol, Moscow, Russia). The assay is based on the use of the allele-specific real-time PCR technique, which makes it possible to detect mutations in the genes associated with resistance to particular drugs. The following gene targets are included in the assay: *rpoB531*, *rpoB526*, *rpoB516* (RIF resistance), *katG315*, *inhA* (INH resistance), *gyrA90*, *gyrA91*, and *gyrA94* (fluoroquinolone resistance). The experiment was carried out on a CFX96 amplifier (BioRad, Hercules, CA, USA). In addition, the WHO-recommended GeneXpert technology and Xpert MTB/RIF reagents (Cepheid, Sunnyvale, CA, USA) were used to detect RIF resistance.

### 2.5. Descriptive Epidemiology Methods

The level and structure of morbidity were assessed by intensive (incidence, prevalence) and extensive (share) indicators. We used information available in the annual statistics forms of the Federal statistical observation in the Omsk region: “Information on the incidence of active tuberculosis”, “Information on patients with tuberculosis”, and “Information on newly diagnosed patients and relapses of tuberculosis”.

### 2.6. Definitions

Newly diagnosed patients were those who did not receive anti-TB treatment or those whose duration of treatment did not exceed one month.

Regarding drug resistance, the following WHO-recommended definitions of *M. tuberculosis* drug resistance types were used. Monoresistance is resistance to only one of the anti-TB drugs used. Polydrug resistance is resistance to two or more anti-TB drugs other than simultaneous resistance to isoniazid and rifampicin. Multi-drug resistance (MDR) is simultaneous resistance to isoniazid and rifampicin. Pre-extensive drug resistance (preXDR) is resistance to rifampicin with or without resistance to isoniazid, in combination with resistance to either a fluoroquinolone or any second-line injectable drug.

The WHO changed the definition of extensive drug resistance in 2020, which is now defined as a combination of MDR with resistance to fluoroquinolones and at least linezolid or bedaquiline [33]. For practical reasons, here we used the previous definition of XDR, i.e., combination of MDR with resistance to fluoroquinolones, and at least one of the three second-line injectables kanamycin, capreomycin, or amikacin.

The requirements of new clinical guidelines for the diagnosis, treatment, and prevention of tuberculosis are planned to be implemented in medical organizations in Russia from the beginning of 2023.

## 3. Results

### 3.1. Characteristics of the Epidemiological Situation

The Omsk region is located in the south-west of Siberia, occupies 141,140 square kilometers, and borders on the Republic of Kazakhstan and other Russian regions in this part of Western Siberia (Tyumen, Novosibirsk, and Tomsk). The Omsk region is one of the provinces of the Siberian Federal District. The district is characterized by the highest levels of incidence, prevalence, and mortality due to TB infection.

In general, 9379 cases of active TB were detected (55.0 cases per 100,000 population) in the Siberian Federal District in 2021. The TB incidence was 1.7 times higher than the national average, which amounted to 45,420 cases (31.1/100,000 population). The largest contribution to the incidence rate was by such provinces as Kemerovo—19.7% (*n* = 1852 cases), Novosibirsk—8.4% (*n* = 1729), and Irkutsk—13.9% (*n* = 1303) (Figure 2 and Figure 3). The Omsk region accounted for 9.8% of all cases in Siberia; the relative incidence rate was 48.1/100,000, which is 14% less than in Siberia. Since 2012, the TB incidence has decreased from 93.9/100,000 by 1.9 times. At the same time, the primary MDR-TB rate increased from 19.2% in 2012 to 26.4% in 2021 (Figure 4).

The TB incidence in women (*n* = 259) was almost three times lower than that in men (*n* = 663): 25.9 versus 75.4 per 100,000, respectively. The TB incidence in the urban population was slightly higher than that in the rural areas and amounted to 49.3 compared to 46.2 per 100,000, respectively.

Among newly diagnosed patients, cases of respiratory tuberculosis prevailed, and the incidence of extrapulmonary tuberculosis was 0.4 cases per 100,000, compared to 1.7/100,000 in Siberia on the whole. The incidence of active TB was 9.0/100,000 in children aged 0–14 years (*n* = 32), and 22/100,000 in adolescents aged 15–17 years (*n* = 12). The incidence was 6.5 higher in adults compared to children.

The continuing high risk of tuberculosis in children who come into contact with sources of TB infection in epidemic foci of tuberculosis is confirmed by a significant TB incidence which, in 2021, increased to 274.2 cases per 100,000 contact children in foci of tuberculosis. The incidence of tuberculosis in adults who were in households or in professional contact with patients with TB infection was somewhat lower and amounted to 147.7 cases per 100,000 contact persons in the epidemic foci. These figures highlight the continued active transmission of *M. tuberculosis*, which contributes to the further spread of TB infection in the region. On average, there were 8.6 contact persons per one TB patient.

Among newly diagnosed patients with pulmonary TB, 35.5% (*n* = 287) were confirmed by sputum smear Ziehl–Neelsen microscopy. All 851 patients with newly diagnosed respiratory tuberculosis in 2021 were examined by the culture method, and 55.2% were culture-positive. Molecular methods were used to directly study sputum samples from 834 patients (98.0%). *M. tuberculosis* DNA was detected in 398 samples (47.7%).

Based on the combined use of all tests (culture, microscopy, or molecular assay), the incidence of respiratory tuberculosis accompanied by the detection of *M. tuberculosis* was 24.5/100,000 (*n* = 470), which amounted to 59.0% of all newly diagnosed patients with respiratory tuberculosis. It should be noted that among newly diagnosed patients with pulmonary tuberculosis, the proportion of destructive forms of tuberculosis, accompanied by lung lesions, was 37.8% (*n* = 288), which is lower than the 45.5% average in Siberia.

### 3.2. Characterization of M. tuberculosis Primary Drug Resistance

All 851 patients with newly diagnosed respiratory tuberculosis in 2021 were examined by the culture method, and 55.2% were culture-positive. A proportion of 94.5% of *M. tuberculosis* isolates (*n* = 444) were tested for drug resistance on liquid and/or solid media. Almost half of the isolates (*n* = 207; 46.6%) were found susceptible to all studied anti-TB drugs. Accordingly, 237 isolates were characterized by resistance to at least one drug (53.4%) (Table 1).

Thirty-three isolates were monoresistant. Of them, 20 were STR-resistant (60.6%), 8 were INH-resistant (24.2%), 3 were OFL-resistant (9.0%), 1 was EMB-resistant, and 1 was PZA-resistant.

Forty-three *M. tuberculosis* isolates were polyresistant. The most frequent combination was resistance to INH + STR (32.5%), followed by INH + STR + EMB (25.6%), INH + PZA (7.0%), INH + STR + OFL (4.6%), INH + STR + PZA (4.6%), and INH + STR + EMB + KAN (4.6%). One isolate was resistant to 6 drugs (RIF, STR, EMB, OFL, KAN, CAP).

MDR profiles were detected in 36.3% (161/444) of the isolates submitted to DST (Table 1 and Table 2). The incidence rate of MDR-TB was 8.4/100,000.

In this study, 161 isolates from newly diagnosed patients were MDR. The choice of effective therapy for such patients was very limited. A significant proportion of the isolates were resistant to most of the first- and second-line drugs. Of the studied MDR isolates, 85.7% (*n* = 138) were resistant to four or more anti-TB drugs. Three isolates were resistant to 11 anti-TB drugs, including the new drugs moxifloxacin and linezolid recommended by the WHO. *M. tuberculosis* isolates from 52 patients who received linezolid treatment were also tested for resistance to this drug and two isolates were found to be linezolid-resistant.

Evaluation of the effectiveness of a course of chemotherapy in patients with pulmonary TB showed that 51.6% of patients newly diagnosed in the previous year (2020) had sputum conversion as demonstrated by sputum microscopy and clinical and radiological methods (*n* = 435).

### 3.3. Drug Resistance-Associated Mutations

Molecular methods were used to directly study sputum samples from 834 patients (98.0%). *M. tuberculosis* DNA was detected in 398 samples (47.7%) and 298 were tested for mutations associated with the development of drug resistance to rifampicin, isoniazid, and fluoroquinolones. Drug resistance mutations were detected in 127 isolates (42.6%) (Table 3). Monoresistance was found in 50 cases (39.4%). The most common INH- resistant mutations were observed in *katG315* (*n* = 26) and only two isolates had mutations in the *inhA* coding region (codon 209). At the same time, rpoB531 mutation was the most frequent among mutations linked to RIF resistance (*n* = 22). Genotypic multidrug resistance was found in 77 samples (60.6%), while INH + RIF resistance was found in 56 (44.1%) isolates and INH + RIF + FQ in 21 isolates (16.5%). Among FQ-resistant mutations, gyrA90 and hyrA94 were the most frequent (Table 3).

## 4. Discussion

The *M. tuberculosis* complex, which is the etiological agent of tuberculosis infection, remains one of the most epidemiologically and clinically significant pathogens. Tuberculosis continues to be a global priority since the continuing upward trend in the prevalence of drug-resistant strains is a major public health problem that threatens the progress made in the fight against tuberculosis [6,9].

The history of anti-TB chemotherapy and the introduction of new drugs in the treatment regime demonstrated that *M. tuberculosis* can acquire resistance to any drug, from classical streptomycin (almost completely phased-out) to the most recent antibiotics such as bedaquiline, and this situation explains the risk of the emergence of new drug-resistant isolates and their epidemic spread [34]. The ability of *M. tuberculosis* to acquire resistance to each particular drug depends on the rate of emergence of spontaneous mutant strains and their physiological fitness. This mutation rate is different for each drug, being higher for pyrazinamide and lower for rifampicin [35,36].

The WHO has strongly recommended drug susceptibility testing of all TB patients to better guide treatment approaches and assess effectiveness. However, the availability of resources in many high-burden countries remains a major challenge [34,37]. In Russia, drug susceptibility testing of both newly diagnosed and chronic patients is mandatory, and more effective methods are being introduced to reduce the time to obtain results and to start or adjust anti-TB treatment [8]. Our study found that almost all TB patients undergo mandatory testing to determine sensitivity to anti-TB drugs and the results obtained provide necessary information for epidemiological surveillance.

The difference found in this study between the three methods used to detect *M. tuberculosis* (microscopy, bacterial culture, and DNA amplification directly from sputum samples) is not unexpected and is explained by their different diagnostic sensitivities. A microscopic smear examination is low-cost, rapid, and easy to perform but it suffers from poor sensitivity. Culture is the gold-standard test for diagnosis of TB. It has high specificity and its sensitivity is considered to be about 100-fold greater than that of smear microscopy. Positive microscopy results have a high predictive value, but relatively low sensitivity necessitates concomitant culturing. On the other hand, certain low-virulence *M. tuberculosis* strains are characterized by slow growth on standard nutrient media. Regarding molecular methods, sensitivity may vary from 90–99% for smear-positive specimens to 66–74% for smear-negative specimens. However, at present, tuberculosis disease cannot be ruled out based on smear-negative and DNA-negative results and culture must always be conducted along with more rapid smear and molecular tests. In addition, molecular tests cannot differentiate between live and non-viable MTBC, so they cannot be used to monitor response to treatment [38,39]. Microscopy-positive cases may be culture-negative and vice versa, but in Russia, more frequently, the culture method, with higher sensitivity and specificity, is used over smear microscopy [16,27,40].

Globally, due to insufficient coverage of patients with drug resistance testing, less than 25% of suspected cases of MDR-TB are detected [37]. According to the Pan American Health Organization in 2017, only 33% of patients in South America underwent drug susceptibility testing, thus leaving about 7000 patients with drug-resistant TB undiagnosed or untreated [41].

There are strong regional differences in the prevalence of MDR-TB in the Siberian Federal District in Russia, indicating a different structure of the circulating *M. tuberculosis* population. For example, in the Tomsk region in 2021, the MDR-TB prevalence was 22.0/100,000 compared to 35.0/100,000 in the Altai region, 47.7/100,000 in the Kemerovo region, and 122.6/100,000 in the Republic of Tyva [28]. In our study, the prevalence of MDR-TB was lower than the average for the Siberian Federal District and amounted to 25.1/100,000. The observed difference can be explained in part by social–economic inequities between the regions and partly by the genetic background of the local human population.

Prolonged anti-TB therapy may lead to the development of drug resistance. Anti-tuberculosis drugs create selective pressure on the *M. tuberculosis* population when spontaneous resistant mutations gradually outnumber susceptible microorganisms and become dominant strains [42]. Monotherapy for tuberculosis leading to drug resistance has been observed since the 1940s, when streptomycin was used as a monotherapy for TB treatment [43,44]. As a result of a long history of their use in TB treatment, INH and RIF resistances are the most commonly detected in circulating isolates from new TB patients [42]. The therapeutic practice of prescribing fluoroquinolones to patients with acute lower respiratory tract infections contributes to the delay in the diagnosis of TB when present and the emergence of fluoroquinolone resistance, which is most dangerous in regions and countries with a high prevalence of TB [45,46]. Treatment of latent TB infection is another potential TB monotherapy practice that has often resulted in isoniazid resistance, especially in countries where bacteriological diagnosis and drug susceptibility testing are not available [47].

In addition to monotherapy and patient non-compliance, underdosing in treatment regimens and poor quality of anti-TB drugs lead to drug resistance. The development of resistance in *M. tuberculosis* begins with monoresistance, followed by acquisition of resistance to two or more anti-tuberculosis drugs through the successive acquisition of resistance mutations [48].

This study presented a detailed description of the structure and spectrum of drug resistance of *M. tuberculosis* isolated from newly diagnosed TB patients in Western Siberia, Russia. The structure of *M. tuberculosis* drug resistance in the Omsk region indicates a significant proportion of strains characterized by multidrug resistance. As a rule, patients infected with susceptible, mono- and polyresistant strains are relatively rapidly and efficiently cured or become microscopy-negative. Patients infected with MDR strains are under dispensary observation for a long time due to ineffective treatment, aggravation, or relapse of the disease. These patients are more likely to transmit drug-resistant strains of *M. tuberculosis* to other members of the population [49]. Therefore, of particular concern are the results indicating a significant proportion of pre-XDR and XDR strains, which accounted for 40.2% and 19.2%, respectively, of all MDR strains.

Negative trends of TB control in the Omsk region are due to the active spread of MDR strains. In particular, the proportion of MDR among chronic TB patients increased from 17.4% in 2012 to 29.8% to 2021. Most alarmingly, the XDR-TB incidence increased 16 times since 2012 and amounted to 1.6/100,000 in 2021, with a long-term average of 1.1/100,000.

The active transmission of *M. tuberculosis* strains with multidrug resistance seriously affects the effectiveness of the treatment of patients and the mortality rate. In our study, 41 patients died of TB. Of them, 18 (43.9%) were under dispensary observation for less than one year. Furthermore, this percentage has almost doubled over the past three years. It should be noted that, in general, in the Siberian Federal District, this indicator is stable and amounted to 26.2%. On the whole, the one-year mortality of newly diagnosed patients in the Omsk region in 2021 was 2.2% of the average annual number of newly diagnosed TB patients. The mortality of TB patients under dispensary observation was 17.8% of the average annual number of all patients with active TB. Among those who died from tuberculosis in 2021, the proportion of microscopy-positive cases was 80.5%.

The first mention of extensively drug-resistant strains appeared in 2006 and so far, such strains have been identified in more than 100 countries around the world [50]. Treatment of XDR-TB patients and sputum conversion in such patients are long and frequently inefficient endeavors, while transmission of drug-resistant strains leads to high mortality rates, which can reach 42–98%, especially in regions with prevalence of HIV coinfection [51]. The proportion of XDR cases among MDR-TB exceeded 10% in Azerbaijan (12.8%), Belarus (11.9%), Latvia (16.0%), Lithuania (24.8%), and Tajikistan (21.0%) [52]. In parts of South Africa, the number of XDR-TB cases increased 10-fold over the past decade [14].

XDR-TB is an almost incurable form of the disease. Currently, in the world, the treatment success rate is 54% for MDR-TB and 30% for XDR-TB [53]. According to the WHO, the main factors contributing to the emergence of XDR-TB are the ineffective treatment of tuberculosis along with the high prevalence of HIV coinfection, the lack of adequate testing to determine drug resistance, and weak control measures to counteract the transmission of *M. tuberculosis* [54].

In 2021, the total number of cases of pre-XDR and XDR-TB was 7 in the USA, 11 in Germany, 37 in the UK, 60 in China, 394 in Uzbekistan, 461 in Kazakhstan, and 7346 in Russia [55]. In our study in the Omsk region, the number of new cases with pre-XDR/XDR-TB was 96. Thirty-one patients were infected with XDR-TB strains and the proportion of XDR among MDR cases was 19.2% (31/161). Unfortunately, at present, commercial drug susceptibility testing methods remain limited for assessing in vitro susceptibility of *M. tuberculosis* isolates to bedaquiline and linezolid [56]. Based on the pre-XDR and XDR data presented, it is not possible to compare the results obtained in different countries that apparently followed different WHO definitions of pre-XDR and XDR-TB in 2021; nevertheless, national surveillance of MDR-TB resistance to anti-TB drugs is important for organizing patient care and planning treatment regimens.

The spread of MDR-TB and XDR-TB strains around the world reflects the insufficiency of the system of global epidemiological surveillance of tuberculosis [55]. XDR-TB significantly limits therapeutic options, and treatment becomes longer, more toxic, more expensive, and less effective, with worse results and higher mortality [57].

Systematic data on the epidemiology of XDR-TB remain limited and most studies focus on molecular studies of selected populations, although knowledge of the prevalence of XDR-TB is relevant for planning the resource support for their examination, informed treatment, and implementation of anti-epidemic measures in the XDR-TB epidemic foci [58]. Surveillance for drug resistance is crucial for identifying and predicting the impact of new empiric prescriptions of anti-TB drugs [56] and the selection of more rational treatment regimens, including new WHO-recommended drugs.

In this study in Western Siberia, we observed the emergence of strains with resistance to newly recommended antibiotics linezolid and moxifloxacin, which is most worrisome. These results clearly show that the transmission of drug-resistant TB among the population of the Omsk region remains relevant and highlights the need to revise preventive and anti-epidemic measures at the regional level to address the problem of further spread of the TB pathogen among the population.

## 5. Conclusions

Despite the decrease in the overall incidence, prevalence, and mortality of TB patients observed in the Omsk region, the epidemiological situation remains extremely alarming. *M. tuberculosis* strains circulating in the southern region of Western Siberia are characterized by a high proportion of multiple, pre-extensive, and extensive drug resistance. In the studied collection of isolates from newly diagnosed patients, the proportion of MDR strains was 67.9%, of which 40.4% were pre-XDR, and 19.2% were XDR (based on the WHO definition used until 2021).

The spectrum of drug resistance is represented by a diverse combination of anti-TB drugs, including second-line drugs (new-generation fluoroquinolone moxifloxacin, linezolid), which significantly limits the possibilities of effective treatment and increases the risk of adverse outcomes related to tuberculosis.

Although the treatment of patients infected with drug-susceptible strains was efficient overall, the treatment of MDR/XDR-TB patients was much less effective, thus leading to the emergence and further spread of MDR *M. tuberculosis* strains; furthermore, strains with additional resistances are acquired during such inefficient treatment.

## Figures and Tables

**Figure 1 microorganisms-11-00425-f001:**
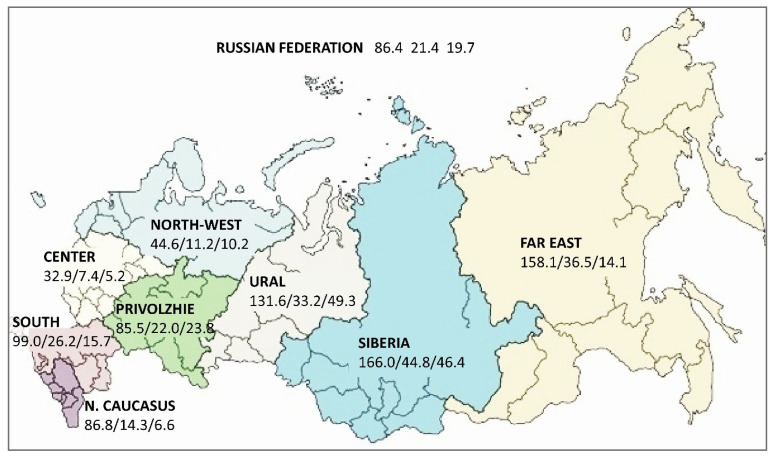
The situation with TB in the Federal districts of the Russian Federation: incidence of TB, MDR-TB, and TB/HIV, per 100,000 population, 2019. Free map: https://en.wikipedia.org/wiki/Federal_districts_of_Russia#/media/File:Map_of_Russian_districts,_2016-07-28.svg (accessed on 15 January 2023).

**Figure 2 microorganisms-11-00425-f002:**
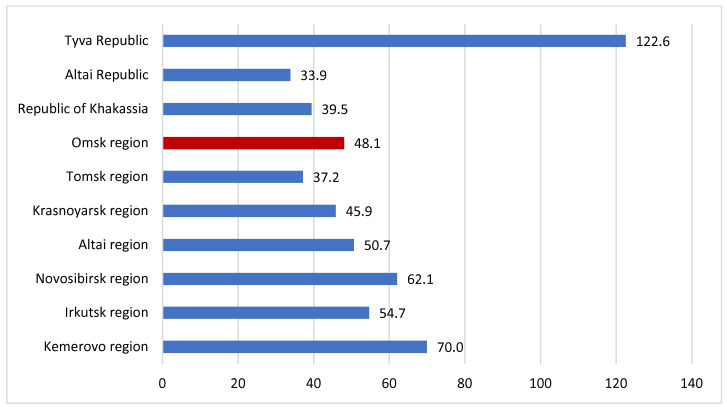
Tuberculosis incidence in the provinces of the Siberian Federal District (2021, per 100,000 population).

**Figure 3 microorganisms-11-00425-f003:**
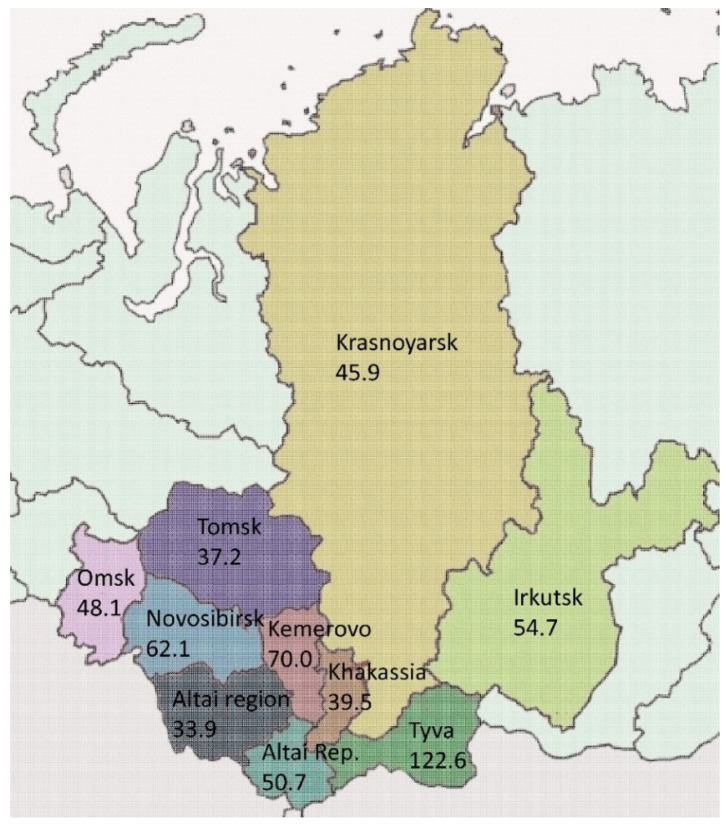
TB incidence in the regions of the Siberian Federal District of Russia. Free map: https://upload.wikimedia.org/wikipedia/commons/thumb/1/1c/Siberian_Federal_District_%28numbered%2C_2018_composition%29.svg/800px-Siberian_Federal_District_%28numbered%2C_2018_composition%29.svg.png (accessed on 15 January 2023).

**Figure 4 microorganisms-11-00425-f004:**
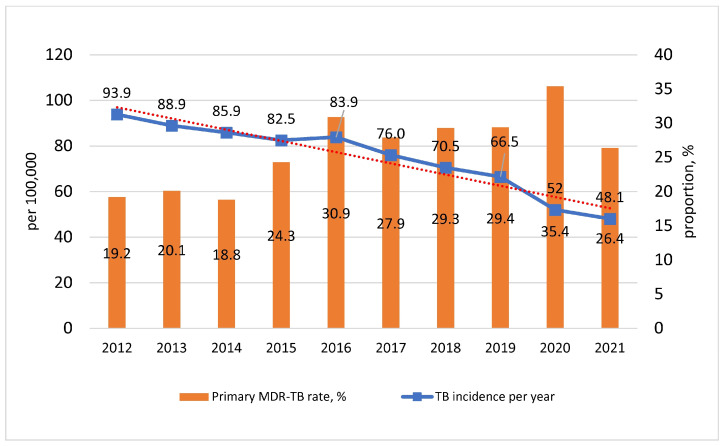
Dynamics of the TB incidence and MDR rate in newly diagnosed TB patients in the Omsk region.

**Table 1 microorganisms-11-00425-t001:** The structure of drug resistance of *M. tuberculosis* isolates isolated from patients with respiratory tuberculosis newly diagnosed in 2021.

Drug Resistance	Number of Isolates	% of All Resistant
Monoresistance	33	13.9
Polyresistance	43	18.2
MDR, including	161	67.9
Pre-XDR	65	27.4
XDR	31	13.0
Total	237	100

**Table 2 microorganisms-11-00425-t002:** Resistance profiles among MDR *M. tuberculosis* isolates from new TB cases in the Omsk region.

MDR Pattern (Resistance to Particular Drugs) ^a^	Number	% (of All MDR)
**MDR (H + R)**	65	40.4
H + R	5	3.1
H + R + S	16	9.9
H + R + S + E	20	12.4
H + R + S + Z	11	6.8
H + R + S + Eto	1	0.6
H + R + S + E + Eto	2	1.2
H + R + S + Z + PAS	1	0.6
H + R + S + E + Z	8	4.9
H + R + S + E + PAS	1	0.6
**Pre-XDR (MDR + Ofx or Km.Am.Cm)**	65	40.4
H + R + Am	2	0.8
H + R + S + Ofx	6	3.7
H + R + S + Km	2	1.2
H + R + S + Am	3	1.8
H + R + S + Cm	1	0.6
H + R + S + E + Ofx	17	10.5
H + R + S + E + Km	6	3.7
H + R + S + Km + Cm	1	0.6
H + R + S + E + Am	3	1.8
H + R + S + Ofx + Z	1	0.6
H + R + S + Ofx + PAS	1	0.6
H + R + E + Km + Eto	1	0.6
H + R + S + Eto + Cm	1	0.6
H + R + S + Z + Am	2	1.2
H + R + S + E + Z + Km	4	2.4
H + R + S + Ofx + Z + PAS	1	0.6
H + R + S + E + Km + Cm	1	0.6
H + R + S + E + Km + Am	1	0.6
H + R + S + E + Ofx + Z	2	1.2
H + R + S + E + Z + Km + Cm	1	0.6
H + R + S + E + Ofx + Z + PAS	1	0.6
H + R + S + E + Z + Km + Eto	1	0.6
H + R + S + E + Km + Eto + PAS	1	0.6
H + R + S + E + Ofx + Z + Lfx	1	0.6
H + R + S + E + Ofx + Eto + Cs + PAS	1	0.6
H + R + S + E + Z + Km + Eto +Cm	1	0.6
H + R + S + E + Z + Km + Am + Cm	2	1.2
**XDR (MDR + Ofx + Km and/or Am, Cm)**	31	19.2
H + R + S + Ofx + Km	2	1.2
H + R + S + Ofx + Cm	1	0.6
H + R + S + E + Ofx + Km	5	3.1
H + R + S + Ofx + Cm + PAS	1	0.6
H + R + S + E + Ofx + Km + Cm	2	1.2
H + R + S + E + Ofx + Km + Pas	1	0.6
H + R + S + E + Ofx + Z + Km	2	1.2
H + R + S + Ofx + Eto + Cm + Pas	1	0.6
H + R + S + Ofx + Z + Am + Cm	1	0.6
H + R + S + E + Ofx + Cm + Lfx	1	0.6
H + R + S + Z + Am + Lfx + Mfx + Lzd	1	0.6
H + R + S + E + Z + Am + Lfx + Mfx	2	1.2
H + R + S + E + Ofx + Z + Am + Cm	1	0.6
H + R + S + E + Ofx + Z + Km + Cm	1	0.6
H + R + S + E + Ofx + Z + Eto + Cm	1	0.6
H + R + S + E + Ofx + Z + Km + Eto	1	0.6
H + R + S + Ofx + Z + Km + Eto + Cm	1	0.6
H + R + S + E + Ofx + Km + Eto + Cm + PAS	1	0.6
H + R + S + E + Ofx + Z + Km + Am + Cm	1	0.6
H + R + S + E + Ofx + Km + Eto + Cm + Cs + PAS	1	0.6
H + R + S + E + Ofx + Z + Km + Am + Cm + Cs + Lzd	1	0.6
H + R + S + E + Ofx + Z + Km + Am + Cm + Lfx + Mfx	2	1.2
Total MDR	161	100.0

^a^ Abbreviations: H, isoniazid; R, rifampicin; E, ethambutol; Z, pyrazinamide; S, streptomycin; Ofl, ofloxacin; Lzd, linezolid; Am, amikacin; Cm, capreomycin; Km, kanamycin; Eto, ethionamid; Cs, cycloserin; Lfx, levofloxacin; Mfx moxifloxacin; PAS, aminosalicylic acid.

**Table 3 microorganisms-11-00425-t003:** Major mutations associated with *M. tuberculosis* resistance to isoniazid, rifampicin, and fluoroquinolones.

Gene	Codon	Anti-TB Drug	Number	% of All Tested
*katG*	315	isoniazid	103	81.1
*inhA*	209	isoniazid	2	1.6
*rpoB*	531	rifampicin	22	17.3
*gyrA*	90	fluoroquinolones	12	9.4
*gyrA*	94	fluoroquinolones	9	7.0

## Data Availability

Data are contained within the article.

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
