# Peer review of "Epidemiology and Drug Resistance Patterns of Mycobacterium tuberculosis in High-Burden Area in Western Siberia, Russia"

_microorganisms, 2023, doi:10.3390/microorganisms11020425_

Round 1

Reviewer 1 Report

The study entitled (Epidemiology and Drug Resistance Patterns of Mycobacterium tuberculosis in High-Burden Area in Western Siberia, Russia) is very interesting and giving alarm about the epidemiological situation in these area. but the are some inquires about the experiment.  

1.       The authors worked on the diagnosis of Mycobacterium tuberculosis complex and Mycobacterium tuberculosis, so the authors should add the paragraph on techniques of diagnosis in the introduction part  

2.       the authors differentiate between the isolated Mycobacterium tuberculosis and other Mycobacterium tuberculosis complex by old biochemical methods. After that, the authors detected two specific genes IS6110 and regX3 for all M. tuberculosis complex. The authors should declare how they confirm and differentiate between M. tuberculosis and other mycobacterium complex (for example. M. bovis).

3.       Give your explanation about the difference between the traditional isolation prevalence and molecular rate for detection of Mycobacterium tuberculosis complex and Mycobacterium tuberculosis

4.       References style should be checked according to the journal style ( Ref. 11, 14, 28,…etc)

Reviewer 2 Report

In the manuscript, the authors present the epidemiological status and drug resistance patterns of infections caused by M. tuberculosis in the Omsk region of Russia. This is an interesting study, primarily because of the very scarce epidemiological information that comes to us from Russia. It is also important that in recent years there are more and more M. tuberculosis infections characterized by severe drug resistance, which is a big problem in treatment.

In my opinion, however, the work needs redrafting. First of all, the title of the work is not compatible with the purpose of the work described by the authors (L. 97-99). This further results in the fact that in the "material and methods" section 2.5 is not related to the described purpose of the work. This, in turn, means that a significant part of the "Results" section should rather be placed in the "discussion" section.

L 165 - "other diagnosic material..." - what??

I think the work should be redrafted.
